# High-Throughput Platform for Efficient Chemical Transfection, Virus Packaging, and Transduction

**DOI:** 10.3390/mi10060387

**Published:** 2019-06-10

**Authors:** Jianxiong Zhang, Yawei Hu, Xiaoqing Wang, Peng Liu, Xiaofang Chen

**Affiliations:** 1Key Laboratory for Biomechanics and Mechanobiology of Ministry of Education, School of Biological Science and Medical Engineering, Beihang University, Beijing 100083, China; 14101035@buaa.edu.cn; 2Department of Biomedical Engineering, School of Medicine, Collaborative Innovation Center for Diagnosis and Treatment of Infectious Diseases, Tsinghua University, Beijing 100084, China; zhangjx1809@163.com (J.Z.); hu-yw17@mails.tsinghua.edu.cn (Y.H.); 3Beijing Advanced Innovation Centre for Biomedical Engineering, Beihang University, Beijing 100083, China

**Keywords:** high throughput gene delivery, transfection, virus packaging, transduction, SMAR-chip

## Abstract

Intracellular gene delivery is normally required to study gene functions. A versatile platform able to perform both chemical transfection and viral transduction to achieve efficient gene modification in most cell types is needed. Here we demonstrated that high throughput chemical transfection, virus packaging, and transduction can be conducted efficiently on our previously developed superhydrophobic microwell array chip (SMAR-chip). A total of 169 chemical transfections were successfully performed on the chip in physically separated microwells through a few simple steps, contributing to the convenience of DNA delivery and media change on the SMAR-chip. Efficiencies comparable to the traditional transfection in multi-well plates (~65%) were achieved while the manual operations were largely reduced. Two transfection procedures, the dry method amenable for the long term storage of the transfection material and the wet method for higher efficiencies were developed. Multiple transfections in a scheduled manner were performed to further increase the transfection efficiencies or deliver multiple genes at different time points. In addition, high throughput virus packaging integrated with target cell transduction were also proved which resulted in a transgene expression efficiency of >70% in NIH 3T3 cells. In summary, the SMAR-chip based high throughput gene delivery is efficient and versatile, which can be used for large scale genetic modifications in a variety of cell types.

## 1. Introduction

In the post-genome era, with the continuous development of genomics, the study of gene function has become a hotspot in life science research. The modification of gene expressions by introducing genetic materials into cells is a common strategy of studying gene function [1]. Usually, a library of genetic materials up to hundreds of thousands needs to be delivered to the cells of interest [2,3]. However, traditional multi-well plate-based techniques rely on manual pipetting where reagents are added the wells one by one, making it difficult to achieve high throughput and to meet the needs of large-scale gene function analysis.

Chemical transfection and viral transduction are two of the most widely used methods for introducing genetic materials into cells. Chemical transfection is easier to conduct, however it obtains high efficiencies only in a few fast dividing cell lines, such as HEK293 cells [4]. On the contrary, virus transduction is more complex, including two steps, virus packaging and target cell infection, but the efficiencies of viral transduction are high in most cell types, including primary cells [5]. Thus, a high throughput platform able to perform both chemical transfection and virus transduction will be ideal due to the effectiveness in most cell types.

Reverse transfection [6,7,8] has been developed for gene delivery in a high throughput manner, where genetic materials are fixed on glass slides to form the transfection islands. Cells are then seeded onto these islands to uptake the genetic materials. Reverse transfection offers the advantages of high throughput and low reagent consumption. However, several intrinsic limitations are associated with this technique, including the possibility of neighboring effects or cross contamination [9] and the inability to perform multiple transfections in a sequential manner which is usually needed to improve the gene delivery efficiencies [10]. Using a similar strategy, reverse transduction has been developed where viruses are fixed on the glass slides to form transduction islands. In addition to the limitations associated with reverse transfection, the viruses have to be prepared before fixing in the conventional culture plates, which is laborious and low throughput [11,12].

Microfluidic devices for gene delivery have been numerously reported, which offer the advantages of high throughput, low reagent consumption, and independently controlled liquid conditions, however most of the studies focused on chemical transfection [13,14] or electroporation [15,16]. Recently, microfluidic systems were employed to increase virus production [17] and improve the target cell transduction efficiencies [18,19] through mass transport-based approaches that overcome the diffusion limitation of the viral particles. However, a high throughput platform integrating virus packaging and transduction has not been reported.

In our previous study, we developed the superhydrophobic microwell array chip (SMAR-chip) for investigating the microenvironmental factors that influence stem cell behavior [20]. Here, we demonstrated that our SMAR-chip can be used as an efficient high throughput platform for gene delivery, which overcomes the limitations of previously reported techniques. Both the chemical transfection and virus transduction can be performed on the chip in a high throughput format, avoiding the well-by-well reagent manipulation. In contrast to reverse transfection, the liquid conditions for individual transfections are completely isolated, reducing the neighboring effects, and the genetic material can be added more than once, enabling multiple transfection in a scheduled manner. More importantly, the whole process from virus packaging to target cell transduction can be done on the chip efficiently, making the platform effective for most cell types. In the future, the SMAR-chip-based gene delivery might be used for large scale gene functional study in a wide range of cell types, including cells resistant to chemical transfections.

## 2. Materials and Methods

### 2.1. Superhydrophobic Microwell Array Chip Fabrication

The chip fabrication was done as previously described [20]. Briefly, a PDMS chip with microwell arrays was fabricated using the standard photolithography technology. A thin layer of polydopamin was coated on the PDMS surface by submerging the chip in a 2-mM dopamine/Tris-HCl solution (pH = 8.5) (Sigma-Aldrich, St. Louis, MO, USA) to enhance cell attachment. The nanoscale superhydrophobic polymer, poly(butyl methacrylate-co-ethylene dimethacrylate) or BMA-EDMA, was synthesized by exposing the premix solution, butyl methacrylate (BMA, 24% wt), ethylene dimethacrylate (EDMA, 16% wt), 1-decanol (60% wt), and 2,2-dimethoxy-2-phenylacetophenone (DMPAP, 1% wt, with respect to monomers) (all from Sigma-Aldrich), to UV light (302 nm, CL-1000 Crosslinker, UVP) for 15 min. A thin layer of Dow Corning 3140 was transferred to the top surface of the PDMS microwell array chip by contact printing, followed by grafting of the BMA-EDMA polymer.

### 2.2. Macroscale Cell Culture

The cells (Plat-E, 293T and NIH-3T3) were cultured in high glucose DMEM, supplemented with 10% FBS, 100 U/mL penicillin, and 100 μg/mL streptomycin (all from Thermal Fisher). All the cells were cultured in a 37 °C incubator with saturate humidity and 5% CO_2_ and passaged every 3–5 days for fewer than 10 passages before being abandoned.

### 2.3. On Chip Cell Culture

Prior to loading cells, the SMARchip was soaked in 75% ethanol for 15 min and exposed to UV light for 2 h. Microwells were first filled with liquid by dripping DMEM onto the chip with a 10-mL stripette, which was held ~20 cm above the chip to expel the air out of the microwells. Cells were then loaded into the microwells by submerging the entire chip into a cell suspension solution for 15 min to allow the cells to settle, followed by aspiration of the excess cell suspension. The chip was then submerged in the culture media overnight for cell attachment.

### 2.4. Macroscale Cell Transfection

293T cells were seeded at a density of 1 × 10^6^/well in six-well plates 1 day before the transfection. The next day, cells were transfected using Lipofectamine^®^ 2000 (Invitrogen, Thermo Fisher Scientific, Waltham, MA, USA) according to the manufacturer’s instruction. The images of the transfected cells were acquired 48 h after the transfection.

### 2.5. On Chip Transfection

293T cells were seeded at a density of ~80 cells/well. After overnight culture, cells in the microwells reached 70–80% confluency. The normal culture medium was aspirated, and the chip was submerged in Opti-MEM with (dry method, see Figure 1C) or without (wet method, see Figure 1C) 0.5% Lipofectamine^®^ 2000 (Invitrogen) for 10 min. Then, the medium was aspirated, and the chip was swept with a cell spreader to form a uniform droplet array. Plasmid solutions (16 μg/mL DNA and 50 mM sucrose in ddH_2_O for the dry method, 16 μg/mL DNA and 1% lipofectamin in Opti-MEM for the wet method) encoding green (pll 3.7, Addgene) or red (constructed by replacing the GFP sequence in pll 3.7 with an RFP sequence) fluorescent protein were spotted on a glass slide in a desired format using a microarray spotter (PersonalArrayer 16, CapitalBio, Beijing, China) that emitted a volume of 30 nL per spot. The glass slide was aligned and sandwiched onto the microwell array with the aid of a custom-made aligner. Then, 4–6 h after the addition of plasmids, the transfection was stopped by submerging the chip in normal culture medium. For multiple-round transfections, three rounds of transfections, each of which lasted 4 h, were consecutively performed without interruption. The images of the transfected cells were acquired 48 h after the last round. For sequential transfections, the two rounds of transfections were performed on day 1 and 2, respectively, with an 18-h interval in between.

### 2.6. Macroscale Virus Packaging and Transduction

The virus packaging cells Plat-E were seeded in six-well plates at the density of 1 × 10^6^ cells per well. The next day, cells were transfected with the pMX-red (Addgene) encoding the red fluorescent protein for 6 h. Viruses were harvested 48 h after the end of the transfection. A total of 1 mL of the virus media supplemented with 8 mg/mL polybrene (Sigma Aldrich) was added to the NIH-3T3 cells (~50% confluent) seeded in the six-well plate. The virus containing media was replaced with fresh media after 6 h and the images of the transducted cells were acquired 48 h later.

### 2.7. On Chip Virus Packaging and Transduction

The virus packaging cells Plat-E were seeded on the SMARchip at a density of ~80 cells/well and incubated overnight. The next day, cells were transfected with the retroviral vector pMX-red (Addgene) encoding the red fluorescent protein for 6 h, using the above-mentioned procedures. The chip was then submerged in normal cell culture media for 10 min to stop the transfection, followed by aspiration of the excess media to form droplets in the microwells. The chip was placed in a petri dish with sterile water to prevent evaporation of the droplets. Virus was produced and accumulated in the droplets for a period of 24 h before aligned and sandwiched onto the chip seeded with the target cells (Figure 4A). Then, 6 h after the assembling, the chip with target cells was submerged in fresh culture media to complete the transduction. The images of the target cells were acquired 48 h later.

### 2.8. Image Acquisition and Analysis

Images of the cells were taken using an inverted microscope (IX83, Olympus) equipped with a charge-coupled device (CCD) camera (iXon3, Andor, Belfast, UK) using a 10× objective and suitable filters for the fluorescence wavelength. The microwell array was scanned with a motorized stage (MS-2000, ASI, Eugene, OR, USA) using a 4× objective. The transfection and the transduction efficiencies were calculated as the number of cells expressing the fluorescent protein genes over the total number of cells in the microwells. The number of fluorescent positive cells was counted manually. All quantification was performed by three researchers blinded to the conditions. The results are reported as mean + standard deviation. The total fluorescent intensity was measured using ImageJ. The images were first thresholded to identify the transfected or infected cells. The fluorescent signal from these cells were calculated as: $S_{c}=S_{ac}-A_{c}\times \frac{S_{w}-S_{ac}}{A_{w}-A_{c}}$, in which *S_c_* is the fluorescent signal from the transfected/transducted cells, *S_ac_* is the total signal from the thresholded cells, *S_w_* is the total signal from the microwell, *A_w_* is the total area of the microwell, and *A_c_* is the total area of the thresholded cells [20]. The results were normalized to the area of the microwell (for the on-chip experiments) or the area of the image (for the macroscale experiments).

## 3. Results

### 3.1. High Throughput Chemical Transfection on the SMAR-Chip

The SMAR-chip with a 13 × 13 microwell array was microfabricated which was composed of a PDMS substrate with the microwells (500 µm in diameter, 200 µm in depth, and 500 µm in pitch, Figure 1A,B) and a layer (~100 µm thick) of pre-synthesized superhydrophobic polymers on the top surface of the PDMS substrate. Droplet arrays were generated on the SMAR-chip due to the repelling effect of the superhydrophobic layer to the aqueous solution, ensuring the formation of isolated liquid conditions in the individual microwells.

We developed two procedures (the dry method and the wet method) to conduct the high-throughput chemical transfection through a few simple steps as shown in Figure 1C. First, the HEK 293 cells were seeded on the chip at a density of ~80 cells/well prior to the transfection. After 12 h of culture, the medium was changed to Opti-MEM and the droplet array was formed by aspirating the excess medium. For the wet method, mixtures of the plasmid solutions and the transfection reagent (Lipofectamine^®^ 2000 here) were spotted on a glass slide in a desired format using a microarray spotter (PersonalArrayer 16, CapitalBio, Beijing, China) that jetted a volume of ~30 nL per spot. The glass slide was then upside-down aligned and sandwiched onto the microwell array with the aid of a custom-made aligner to deliver the DNA to the cells. After 4–6 h of incubation, the transfection was stopped by submerging the chip in normal culture medium. The DNA-Lipofectamine^®^ 2000 mixtures should be delivered to the cells immediately after spotting to avoid evaporation.

While for the dry method, the transfection reagent (Lipofectamine^®^ 2000) was added to the Opti-MEM and delivered to the cells during the submerging step (step 2). The glass slide with the spotted plasmid solution can be dried and preserved for up to 15 months [6], making it more convenient for the end users (step 4). 50 mM sucrose was added to the DNA solution which functions as a scaffold to protect the DNA molecule from absorbing to the glass slide.

As a proof-of-concept of the high-throughput chemical transfection methods, plasmid solutions encoding green or red fluorescent proteins were spotted on a glass slide in an alternate line format. In the dry method, all the slides were spotted and dried for more than 24 h. As the DNA delivery was convenient and repeatable, three rounds of transfections were consecutively performed to improve the efficiency (Figure 2A). The images of the transfected cells were acquired 48 h after the last round. We observed that the transfection efficiencies significantly improved from 35% ± 15% in the first attempt to 65% ± 16% in the second and 69% ± 16% in the third and the total fluorescent signal of the transfected cells also increased from the first to the third rounds. By contrast, the wet method facilitated the formation of the DNA-liposome complex due to the premixing step, thus one round of transfection resulted in an efficiency of 62% ± 16%, comparable to conventional method conducted in cell culture plates (63% ± 8.3%, Figure 2C). As we expected, no cross contamination was observed under both the dry and wet methods even after three rounds of transfections (Figure 2D). The initial cell number must be carefully controlled in order to achieve higher efficiencies, as too many cells will rapidly reach the capacity of microwells and too few cells will not grow well. A density of ~80 cells/well will achieve the optimal results.

### 3.2. Sequential Gene Transfection

We also performed sequential GFP and RFP transfections with different delivery patterns on day 1 and day 2, respectively, with an 18-h interval in between. As illustrated by the upper panel of Figure 3A, plasmids encoding RFP were delivered to the cells in an alternate line format on day 1 and plasmids encoding GFP were delivered to all the cells 18 h after the end of the first transfection. On day 2 (prior to the addition of GFP plasmids), cells expressing RFP signals were seen in alternate lines while no GFP signals were detected, same as the plasmid delivery pattern. On day 3 and day 4 (24 and 48 h after the delivery of GFP plasmids), GFP expression cells were observed in all the microwells while the expression of RFP was still in the alternate line manner same as that in day 1 (Figure 3A, lower panel). In addition, efficient transfections of both the RFP and GFP plasmids were achieved throughout the chip as shown by the merged fluorescent micrograph of the 13 × 13 microwell array in Figure 3B. These results demonstrated that both of the genes were successfully transfected without cross contamination, proving the feasibility of conducting schedule-based gene transfections on the chip.

### 3.3. Virus Packaging and Transduction

Although the chemical transfection is easy to conduct, its efficiency is low for most of the primary cells [21]. Exploiting the viral infection pathway, the method of viral transduction is efficient for most of the cell types [22] and has been widely used in studies of stem cells and primary cells. Typical virus transduction includes two steps: (1) virus packaging by the packaging cells, and (2) transduction of the target cells by adding the virus containing medium. These two steps can all be performed on the SMAR-chip in a high throughput manner as illustrated in Figure 4A. The retroviral packaging cells Plat-E were seeded on the SMAR-chip and transfected with the viral vector encoding RFP as described above. In the following 48 h, viruses were released into the droplets in each microwell. To infect the target cells, the chip with the viruses was upside-down aligned and sandwiched with the chip seeded with NIH-3T3 cells, which showed very low transfection efficiencies (22% using Lipofectamine^®^ 2000) in published data [23,24]. As shown in Figure 4B, the expression of RFP was observed in 76% ± 16% of the NIH-3T3 cells, demonstrating successful virus packaging and transduction.

## 4. Discussion

The SMAR-chip-based gene delivery is convenient and versatile which overcomes the limitations of the conventional multi-well plate-based techniques and the reverse transfection. Here, 169 transfections were done simultaneously in physically separated microwells on the SMAR-chip. The throughput can be further increased by fabricating larger chips with more microwells. The labor consuming well-by-well operations are avoided since media exchange in all the microwells is done by submerging the chip in fresh media for 10 min followed by aspirating the excess media. The genetic materials are delivered to the microwells by sandwiching the DNA array with the SMAR-chip. In the future, DNA can be added to the microwells through centrifuge using our previously reported nanoliter centrifugal liquid dispenser to further simplify the whole procedure [25]. In contrast to reverse transfection, the formation of droplets generates isolated liquid conditions in individual microwells, reducing the neighboring effects. The capability to perform multiple transfection in a scheduled manner can further increase the efficiency and flexibility of transgene expression.

In addition, the dry and the wet methods have distinct advantages, and both achieve efficiencies higher than 60%, similar to the conventional transfection in cell culture plates. The dry method is more convenient to the end users since the glass slides with spotted DNA arrays can be preserved for a long time, thus not requiring the availability of the microarray spotter. While, the wet method has higher transfection efficiency as the premixing of DNA with the transfection reagent (Lipofectamine^®^ 2000) facilitates the formation of the DNA-liposome complex.

More importantly, on chip high-throughput virus production combined with in situ target cell infection was achieved. As a proof of concept, transduction of the NIH 3T3 cells results in expression of the transgene in about 76% of the target cells, much higher than the previously reported 22% for chemical transfections [23,24]. These results suggest that the SMAR-chip-based high-throughput gene delivery will be efficient for a broad range of cell types, including those resistant to chemical transfections.

In summary, the SMAR-chip can be used as an efficient platform for high throughput gene delivery through chemical transfection or virus transduction. In the future, a suitable method can be chosen according to the cell type and the experimental design. In addition, the SMAR-chip technology can be used in other high-throughput cell microarray-based assays, such as drug screening and the analysis of factors that influence cell behavior, owning to the convenience of reagent delivery and the isolated media conditions in individual microwells.

## Figures and Tables

**Figure 1 micromachines-10-00387-f001:**
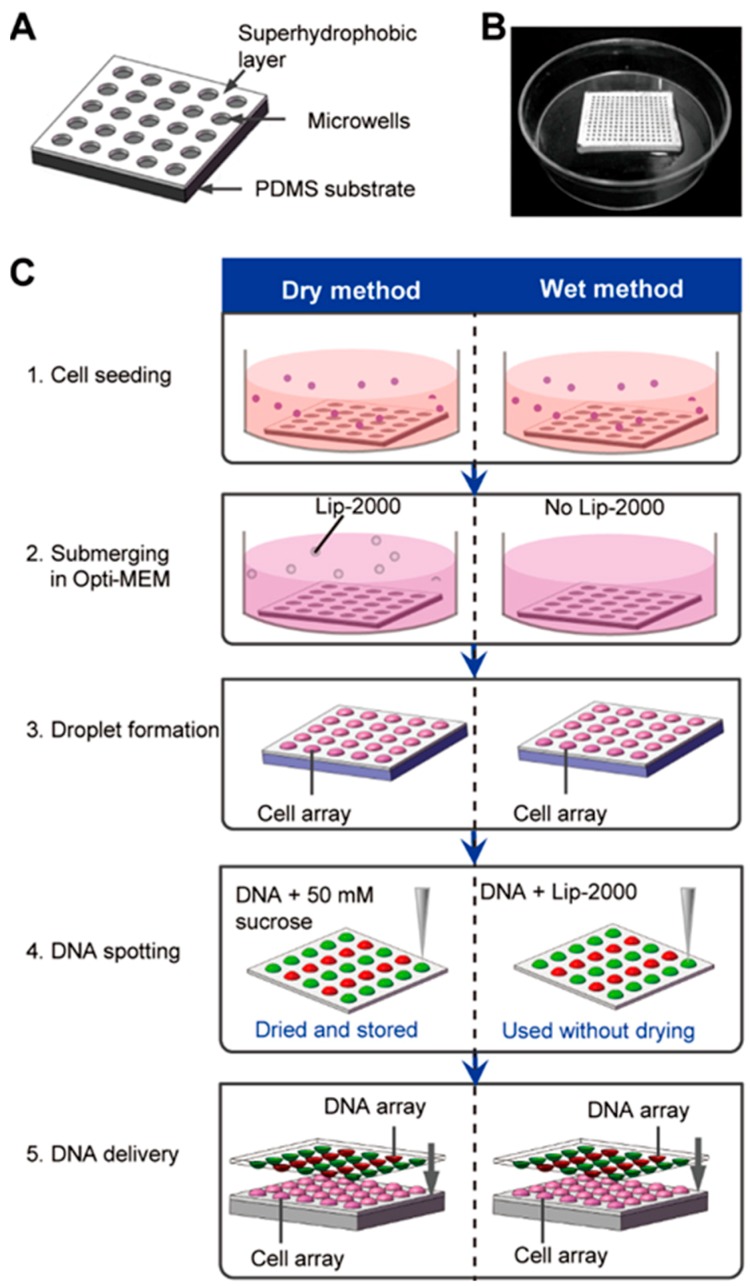
Schematic of the high-throughput transfection on the chip. (**A**) Structure diagram of the superhydrophobic microwell array chip (SMAR-chip). (**B**) Macroscale image of a SMAR-chip with a 13 × 13 microwell array. (**C**) Procedure of the on-chip high-throughput transfection. For the dry method (left), the mixture of DNA and 50 mM sucrose was spotted on the glass slide which can be dried and stored until the time of use, while for the wet method (right) the mixture of DNA and Lipofectamin^®^ 2000 (Lip-2000) was spotted and used immediately without drying.

**Figure 2 micromachines-10-00387-f002:**
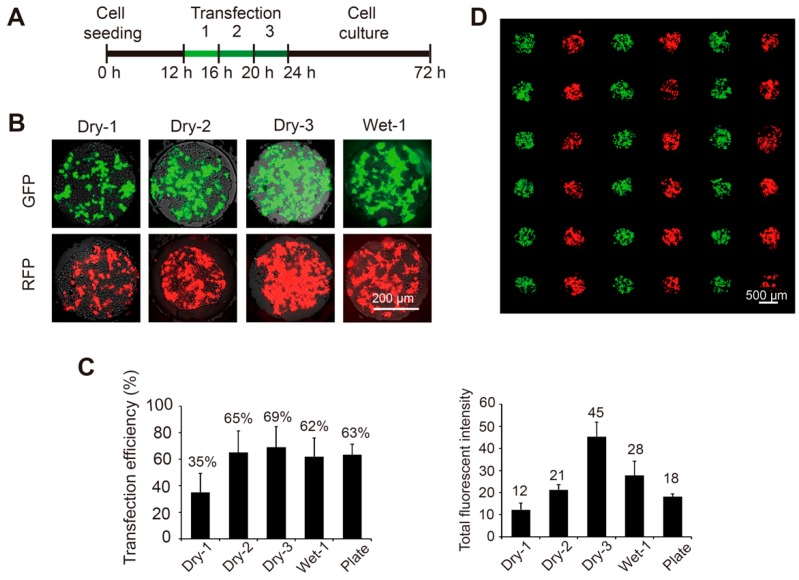
Transfection of plasmids encoding the green and red fluorescent protein (GFP and RFP) genes into 293T cells on the SMAR-chip. (**A**) Timeline of the on chip transfection (dry method). 293T cells were seeded in the microwells. After a 12-h recovery, one to three rounds of transfections were conducted consecutively, followed by two days of culture. (**B**) Typical fluorescent images of the cells transfected with GFP and RFP (upper panel) using the dry method for one (Dry-1), two (Dry-2), and three (Dry-3) rounds or using the wet method for one round (Wet-1). (**C**) Left panel: the transfection efficiencies (left panel) which are calculated as the number of cells with detectable fluorescent signal over the total number of cells in the microwells. Right panel: the total fluorescent signal from the transfected cells data are represented as the mean + standard deviation (*n* = 20). (**D**) Merged fluorescent image of 293T cells transfected with GFP and RFP, indicating efficient transfection without cross contamination.

**Figure 3 micromachines-10-00387-f003:**
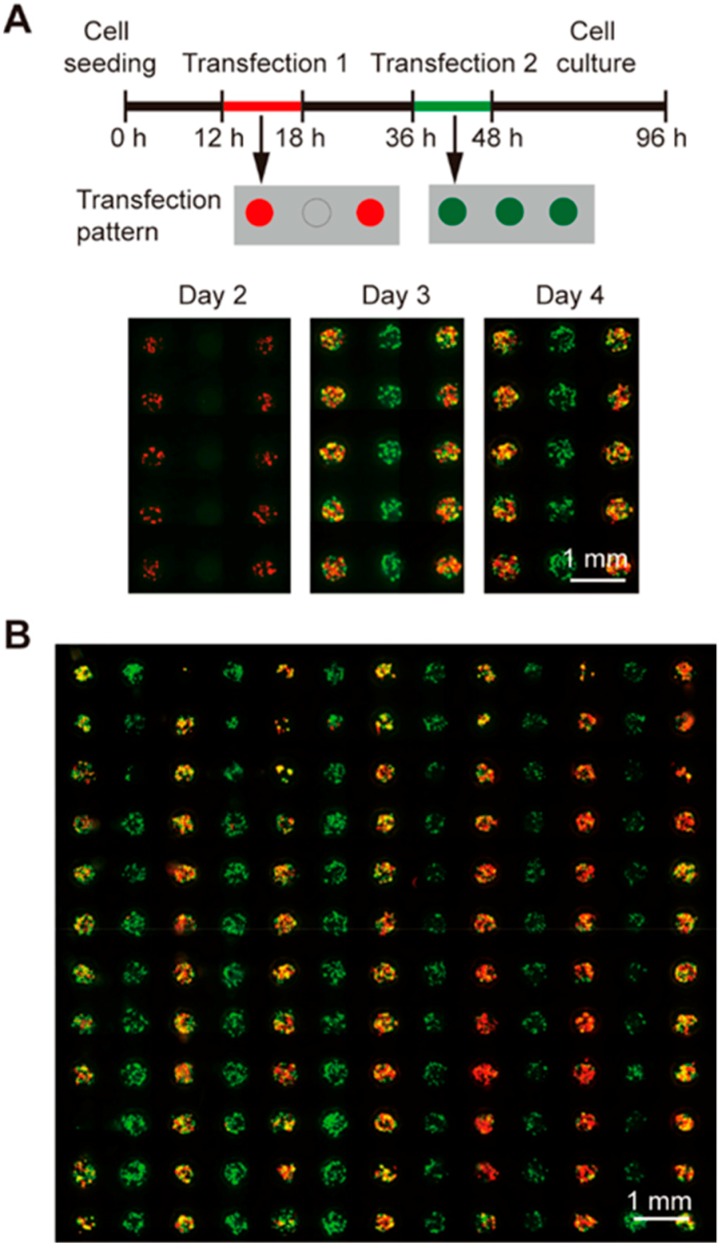
Sequential transfections of different plasmids encoding GFP and RFP into 293T cells on the SMAR-chip. (**A**) The transfection protocol is illustrated in the upper panel. RFP was transfected into the cells in alternate lines 12 h after cell seeding, and GFP was transfected to all the cells 18 h after the RFP transfection. Fluorescent images shown in the lower panel were taken sequentially at day 2, 3, and 4. (**B**) Merged fluorescent micrograph of 293T cells sequentially transfected with green and red fluorescent proteins. (scale bars, 1 mm).

**Figure 4 micromachines-10-00387-f004:**
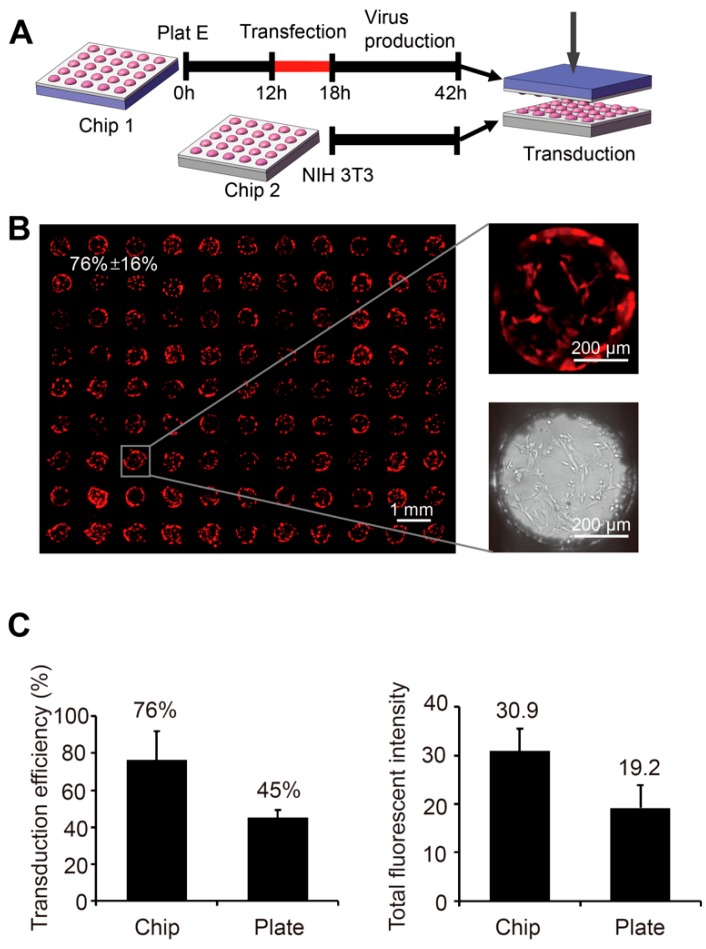
Gene delivery through virus transduction on the SMAR-chip. (**A**) The schematic of virus packaging and target cell transduction. The packaging cells (Plat E) were seeded on the SMARchip and transfected with the virus plasmids encoding RFP. Then, 48 h after the transfection, the virus containing chip was turned upside down and aligned with the chip seeded with the target cells (NIH-3T3). (**B**) The fluorescent images of the NIH-3T3 cells two days after virus transduction. (**C**) Quantification of the transduction efficiency (left panel) and the total fluorescent intensity (right panel) of the on chip transduction (Chip) and conventional transduction in six-well plates (Plate).

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
