# Peer review of "High-Throughput Platform for Efficient Chemical Transfection, Virus Packaging, and Transduction"

_micromachines, 2019, doi:10.3390/mi10060387_

Round 1
Reviewer 1 Report
The manuscript by Zhang et al describes a high throughput chip-based platform for gene delivery, based on the previously published by the same authors superhydrophobic microwell array chip (SMAR-chip, Zhang, P.F.; et al. Lab Chip. 2016, 16, 2996-3006.). The presented device and its specific application is very innovative and overcomes the limitations of previously reported techniques for high throughput transfection and transduction.
The manuscript is well written overall and Introduction covers the most important topics appropriately. Methods are properly presented, are detailed and to the point. Results are well laid out with very good and clear figures that support the text in a logical manner.
Discussion is based on the current work in the field and outlines advantages of the SMAR technology in a balanced fashion. References are up to date and sufficient in my opinion.
I have only a few minor suggestions for further improvement:
1) Figures 2, 3, 4 – no quantitative data are provided in the form of charts despite the fact that text describes quantitative data. This should be addressed
2) In the discussion section authors could elaborate if the current SMAR technology could be modified to be employed in high-throughput microfluidic arrays to simplify and automate fluid/reagent delivery etc
Author Response
1) Figures 2, 3, 4 – no quantitative data are provided in the form of charts despite the fact that text describes quantitative data. This should be addressed.
Following the reviewer’s suggestion, we added quantitative data in the figures. In Figure 2, we quantified the transfection efficiencies of the dry and wet methods as shown in panel C. In figure 4, we compared the efficiencies of on chip infection and conventional infection in 6-well plates as shown in panel C.
2) In the discussion section authors could elaborate if the current SMAR technology could be modified to be employed in high-throughput microfluidic arrays to simplify and automate fluid/reagent delivery etc.
This is a great suggestion. Our SMAR-chip technology can be used in other high-throughput microfluidic arrays, the sentences “In addition, the SMAR-chip technology can be used in other high-throughput cell microarray-based assays, such as drug screening and the analysis of factors that influence cell behavior, owning to the convenience of reagent delivery and the isolated media conditions in individual microwells” are added to the last paragraph of the discussion section.
Reviewer 2 Report
The authors developed microcell array chip for a large scale gene transfection. The approach is interesting and seems to have a potential for future practical usage. However, evaluation methods should be improved for clarifying the advantage of their system. Please find specific comments below.
1) l. 128 It is better to clarify the methods to calculate the number of fluorescent positive cells. Did they use software or count manually? How was the bias in the counting managed?
2) In the evaluation of transfection efficiency, total fluorescence intensity, as well as the percentage of transfected cells, is important. The total fluorescence intensity should be shown.
3) Only a little information was provided regarding the comparison with conventional transfection procedure. Bar charts should be shown to clarify the total fluorescence intensity, and the percentage of transfected cells of the newly developed system and conventional transfection methods.
4) In figure 1B, 4B, GFP- or RFP-positive cells were clustered, leaving other area untransfected. Why did the area remain untransfected? Did the area have cells? Was DNA
added to the area? Was the DNA distributed uniformly in the well? Addition of fluorescence labeled DNA would be helpful to answer this question.
5) It is improper to claim that this method is helpful to reduce the cross contamination between wells just from Figure 2C or similar results, because fluorescence-based method is too insensitive for detecting a contamination. Even in conventional transfection, the amount of DNA added to unintended wells seems to be very little, presumably far less than 1%. Such small contamination cannot be detected in the present method. Other methods should be performed, for example, quantitative real time PCR. This issue is critical to clarify the advantage of the newly developed system.
Author Response
1) l. 128 It is better to clarify the methods to calculate the number of fluorescent positive cells. Did they use software or count manually? How was the bias in the counting managed?
The number of fluorescent positive cells was counted manually. All quantification was performed by three researchers blinded to the conditions to avoid bias. The sentences “The number of fluorescent positive cells was counted manually. All quantification was performed by three researchers blinded to the conditions. The results are reported as mean +standard deviation.” were added to line 141 on page 3.
2) In the evaluation of transfection efficiency, total fluorescence intensity, as well as the percentage of transfected cells, is important. The total fluorescence intensity should be shown.
This is a great suggestion. We added the quantification of the total fluorescence in Figure 2C and Figure 4C. The detailed procedure to quantify the fluorescent signal was added to image acquisition and analysis in the materials and methods section on page 3.
3) Only a little information was provided regarding the comparison with conventional transfection procedure. Bar charts should be shown to clarify the total fluorescence intensity, and the percentage of transfected cells of the newly developed system and conventional transfection methods.
Following the reviewer’s suggestion, bar charts for the comparison between the on-chip and the conventional procedures were added in Figure 2 (chemical transfection) and Figure 4 (virus infection). Both the percentages of transfected cells and the total fluorescence intensities were shown. The experimental procedure of the conventional transfection and transduction were added to the materials and methods section on page 3.
4) In figure 1B, 4B, GFP- or RFP-positive cells were clustered, leaving other area untransfected. Why did the area remain untransfected? Did the area have cells? Was DNA added to the area? Was the DNA distributed uniformly in the well? Addition of fluorescence labeled DNA would be helpful to answer this question.
The reasons why some of the area was untransfected are: 1) the cells were not 100% confluent thus some of the area has no cells, 2) some of the cells (about 30%) were not transfected which is normal in all the DNA transfection experiments. We believe that the DNA was added to the microwells and the DNA distribution was uniform. We validated the reagent delivery method in our previous study by Zhang et al (Reference 20 in the manuscript). We observed that the fluorescent dye was added to the microwells and the fluorescent intensity in the microwells was uniform as shown in the following figure. Since the reagent delivery method used in this study was the same as that in our previous study, we believe DNA was uniformly distributed in the microwells.
5) It is improper to claim that this method is helpful to reduce the cross contamination between wells just from Figure 2C or similar results, because fluorescence-based method is too insensitive for detecting a contamination. Even in conventional transfection, the amount of DNA added to unintended wells seems to be very little, presumably far less than 1%. Such small contamination cannot be detected in the present method. Other methods should be performed, for example, quantitative real time PCR. This issue is critical to clarify the advantage of the newly developed system.
In our previous study by Zhang et al, we verified the reagent delivery method. The results are shown in the following figure. As we can see, each reagent spot made contact with one microwell whereas the merging of neighboring droplets was not observed. Red and green fluorescent dyes printed in alternate lines were successfully added into the microwells while fluorescent signals from the unintended microwells were not detected (please see the figure in the uploaded file). As the reviewer pointed out, very small amount of contamination may exist which cannot be detected by the current method. However, this small amount of DNA is not enough for transfecting any cells. Our method reduced the “neighboring effects” or cross contamination of the reverse transfection where all the cells were submerged in the same media, leading to the cellular communication and DNA diffusion through the medium (reference 9 in the manuscript). By contrast, the isolated media conditions were generated in individual microwell on our chip, preventing cell communication and the diffusion of DNA among the neighboring microwells. In addition, since the main point of the current study is not whether our method can reduce cross contamination in comparison to the conventional microplate or not, we believe further demonstration is not that necessary. To clarify this point, we made the following changes to the manuscript:
1) We deleted the sentence “thus avoiding the cross contamination” in line 157, page 4.
2) We changed the sentences “In contrast to the reverse transfection, the liquid conditions for individual transfections are completely isolated, eliminating the possibility of cross contaminating” to “In contrast to the reverse transfection, the liquid conditions for individual transfections are completely isolated, reducing the neighboring effects” in line 70, page 2.
3) We changed the sentences “In contrast to reverse transfection, the formation of droplets generates isolated liquid conditions in individual microwells, eliminating the possibility of cross contamination” to “In contrast to reverse transfection, the formation of droplets generates isolated liquid conditions in individual microwells, reducing the neighboring effects” in line 264, page 8.

Round 2
Reviewer 2 Report
The answers from authors are reasonable. The manuscript is now acceptable.